# Perceptual harmony in judgments of group prototypicality and intragroup respect

**Joshua D. Wright** [1]*, **L. James Climenhage**[1], **Michael T. Schmitt**[1], **Nyla R. Branscombe**[2]

**1** Department of Psychology, Simon Fraser University, Burnaby, British Columbia, Canada, **2** Department of Psychology, University of Kansas, Lawrence, Kansas, United States of America

* jwright6@sjcny.edu

**Data Availability Statement:** Data are available on the Open Science Framework: https://osf.io/wxm5r/.

**Funding:** This research was supported through a Social Sciences and Humanities Research Council

## Abstract

We test common sense psychology of intragroup relations whereby people assume that intragroup respect and ingroup prototypicality are positively related. In Study 1a, participants rated a group member as more prototypical if they learned that group member was highly respected rather than disrespected. In Study 1b, participants rated a group member as more respected by other group members if they learned that group member was prototypical rather than unprototypical. As a commonsense psychology of *groups*, we reasoned that the perceived relationship between prototypicality and intragroup respect would be stronger for cohesive groups compared to incohesive groups. The effect of intragroup respect on perceptions of prototypicality (Study 2a & 2c) and the effect of prototypicality on perceptions of intragroup respect (Study 2b) were generally stronger for participants considering cohesive groups relative to incohesive groups. However, the interaction effect of prototypicality and group cohesion on intragroup respect did fail to replicate in Study 2d. In Studies 3, 4a, and 4b we manipulated the relationship between prototypicality and intragroup respect and found that when these variables were in perceptual harmony participants perceived groups as more cohesive. The results of eight out of nine studies conducted are consistent with the prediction that people make inferences about intragroup respect, prototypicality, and group cohesion in a manner that maintains perceptual harmony.

## Introduction

A folk theory in psychology represents "a framework of concepts, roughly adequate to the demands of everyday life", which is used to understand, explain, and predict phenomena [1, p. 225]. Such folk theories or "common-sense psychology" reflect the "natural, intuitive, common-sense capacity" of humans to grasp the complex interplay between self, other, and environment [2, p. 5]. Common sense psychology can contain essential elements of formal scientific theories, can form the basis of scientific theories, and can guide accurate inferences for lay people. Heider [2, p.58] suggested that, "The world we perceive has to be consistent, and the equivocal stimuli, even ordinal stimuli, will give rise to percepts that fit together and produce an integrated picture." This consistency principle, or balance, indicates a uniform view of elements that are in unit relations with each other [3]. Unit relations refers to any two

of Canada (SSHRC) grant to the third author. The funders had no role in study design, data collection and analysis, decision to publish, or preparation of the manuscript.

**Competing interests:** The authors have declared that no competing interests exist.

entities that are perceived as belonging together [2], whether through perceived similarity, causal relation, or proximity [4].

The principle of balance suggests that individuals prefer cases in which relations among entities fit together harmoniously [2]. Individuals can use the balance schema as an intuitive inferential rule to quickly make sense of social relationships [5]. When a state of balance is absent from a set of relations between entities, this may result in a tendency to remove the state of imbalance. As an example, assume that person $p$ respects person $o$. This reflects a positive sentiment relation. Now assume that $o$ told a lie. This reflects a positive unit relation between $o$ and lying. Finally, assume $p$ dislikes lying (a negative sentiment relation). The example reflects a state of imbalance among the relations between entities and would induce change to reinstate and maintain affective and cognitive consistency [2]. This could be accomplished by altering the sentiment relation between $p$ and $o$ or by altering the sentiment relation between $p$ and the act of lying. Either would induce balance. Regarding these triad relations, Heider states,

> *"A triad is balanced when all three of the relations are positive or when two of the relations are negative and one is positive. Imbalance occurs when two of the relations are positive and one is negative (p. 202)"*

The balance schema fundamentally explains predictions that individuals make within interpersonal relations, such as predicting that my friend's friend is someone whom I would also like, or that my friend's enemy is someone whom I would not like [2]. We suggest that the balance schema also guide's intuitive judgements about groups that may be functional while navigating interactions with groups. For example, if intragroup respect and prototypicality were inferred from one another then individuals could use this information when navigating new membership in a group. New members to a group might observe who the highly respected members are and then infer that these members most embody the group identity and subsequently that their values and behaviors should be emulated.

## Application of the principle of balance to groups

We begin with the simplest case of assuming a collective of individuals who cohesively form a group, which reflects a positive unit relation between group members ($o$) and the group category ($y$). As displayed in panel A of Fig 1, perceptions are in a balanced state when person $p$ is perceived to be prototypical of group $y$ (a positive unit relation between $p$ and $y$), and $p$ is perceived as respected by other group members, $o$ (positive sentiment relation). As displayed in panel B of Fig 1, balance would also occur in the case of $p$ being perceived as unprototypical of group y (negative unit relation) and *not* respected (negative sentiment relation).

Two imbalanced cases are depicted in panels C and D of Fig 1. In panel C, $p$ is perceived to be prototypical of group $y$, but $p$ is perceived as not respected by other group members ($o$). This leaves two positive relations and one negative relation—a state of imbalance. Likewise, in panel D, $p$ is perceived to be unprototypical of group $y$, but $p$ is perceived as respected by other group members ($o$).

Cohesion reflects the extent that individual members are bonded together [6]. When groups are perceived as cohesive, perceptual accentuation—exaggerated perceptions of ingroup homogeneity [7]—can occur. Similarly, it is the positive relation between group members ($o$) and the category ($y$) that can create additional pressure toward perceiving prototypicality and intragroup respect as positively related. In the case that group cohesion is very low, and that relation does not exist, there is no pressure to perceive consonance between

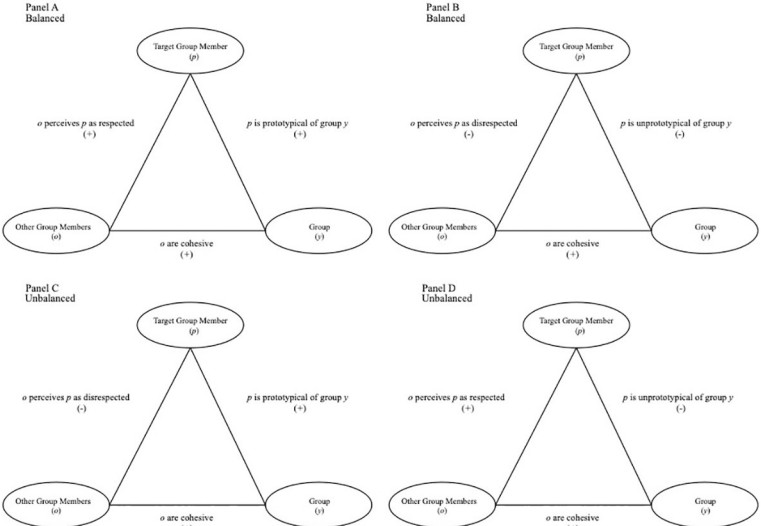

**Fig 1. Balanced and imbalanced states under the condition of group cohesion.** Panels A and B reflect states of balance. Panels C and D reflect states of imbalance. A "+" indicates a positive relation and a "-"indicates a negative relation.

perceptions of prototypicality and respect. When there is no relationship between the category and group members, the relative levels of prototypicality and respect for any one group member are not relevant to perceptual harmony.

## The current studies

Social categorization theory has identified a positive relationship between intragroup respect and member prototypicality—that is individuals who are more prototypical of their groups are also more respected by their fellow group members [8, 9]. However, no one has investigated *whether lay people hold a similar folk theory that guides their perceptions of groups and individual members of groups*. The balance schema may serve a functional purpose by guiding individual prediction and action within groups. We suggest that individuals hold a lay theory that matches Heider's concept of balance and we examine this proposition in nine studies. In Studies 1a and 1b, we evaluate whether lay individuals make inferences about a group member's prototypicality and intragroup respect in a pattern that reflects balance. In Studies 2a and 2b we evaluate whether these inferences depend upon the cohesiveness of a group. In Studies 2c and 2d we attempt to replicate the findings from 2a and 2b in high powered and preregistered designs. In Studies 3 and 4a, we examine whether individuals make inferences about group cohesion on the basis of perceived relationships between prototypicality and respect and in Study 4b we attempt to replicate Study 4a in a high powered and preregistered design. Finally, we compute a *p*-curve of the reported studies to demonstrate their evidential value. Data collection was completed between 2008 and 2010 for all studies with the exception of 2c, 2d, and 4b, for which data were collected in 2020.

## Study 1a and 1b

We first examine a simple case of Heider's principle of balance. Prototypes exist as group member generalizations that reflect general patterns of perceived positive ingroup traits [10]. Thus, we begin by examining whether lay individuals infer the prototypicality of members of groups from knowledge of the extent to which a member is respected by other ingroup

members (1a) and infer intragroup respect of members of groups from knowledge of how prototypical a group member is (1b). Within Heider's [2] conceptualization of balance, a balanced state would be one in which the perceiver views prototypical members as respected members and views non-prototypical members as non-respected members. Additionally, we examine whether these lay inferences occur regardless of whether groups are voluntarily chosen (e.g., a club) or compulsory (e.g., a required class). Throughout this manuscript, all measures, manipulations, and exclusions are disclosed for all studies and we provide a sensitivity power analysis using G*Power [11] for all statistical tests conducted. No data collection was continued after data analysis.

## Study 1a methods

All studies reported received ethics approval from the Research Ethics Board at Simon Fraser University.

**Participants.** Eighty students (49 women, 31 men) from Simon Fraser University participated in the current study in exchange for course credit. Participants ranged in age from 18 to 36 years ($M$ = 19.74, $SD$ = 2.45). The sample included participants identifying as East Asian (n = 38), White (n = 19), South Asian (n = 12), mixed ethnicities (n = 6) and other (n = 5).

**Design and procedure.** Participants were randomly assigned to one of four experimental conditions, with 20 participants in each condition. Sample size was chosen based upon rules of thumb at the time of data collection, and pragmatic constraints on data collection from a subject pool, which included maximum allotments determined by the department. Participants read a description of "M", in which M's group was either framed as compulsory ("M enrolled in a grade 12 math class; a requirement for graduation") or voluntary ("M joined a math club, an extra-curricular activity at his high school") and in which "M" was framed as respected ("M is a very respected member of his math [class/club]") or disrespected ("M is a very disrespected member of his math [class/club]"). Thus, the study incorporated a 2 (group type: compulsory vs. voluntary) x 2 (target type: respected vs. disrespected) design.

After reading the short vignette, participants were asked, "To what extent would you expect M to be a prototypical or representative member of his math [class/club]?" Participants rated target prototypicality on a 7-point scale ranging from 1 (very unprototypical) to 7 (very prototypical), followed by demographic questions.

## Study 1a results

A 2 (group type) x 2 (target type) ANOVA was conducted with inferred target prototypicality as the dependent variable. There was no effect of the interaction between group type and target type, $F$ (1, 76) = .58, $p$ = .45, and no main effect of group type, $F$ (1, 76) = .07, $p$ = .80. There was a statistically significant and large effect of target on inferred prototypicality, $F$ (1, 76) = 44.83, $p < .001$, $g$ = 1.52. Participants inferred higher prototypicality of the target when the target was framed as a respected member of his group ($M$ = 4.78, $SD$ = 1.35) rather than a disrespected member of his group ($M$ = 2.80, $SD$ = 1.26). Fig 2 displays these results with 95% confidence intervals for the group means. A sensitivity power analysis suggested that we had 80% power to detect effects as small as $d$ = .76 for the interaction between group type and target type and $d$ = .64 for the main effects.

## Study 1b methods

**Participants.** Seventy-eight students (46 women, 32 men) from Simon Fraser University participated in the current study in exchange for course credit. Participants ranged in age from 18 to 23 years ($M$ = 19.15, $SD$ = 1.27). The sample included participants identifying as East

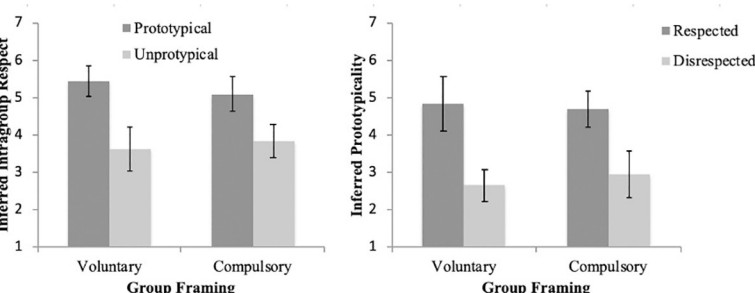

**Fig 2. Inferred intragroup respect and inferred prototypicality as a function of target prototypicality and target intragroup respect.**

Asian (n = 41), White (n = 23), South Asian (n = 4), mixed ethnicities (n = 2) and other (n = 8).

**Design and procedure.** This study was identical to 1a except that "M" was framed as either prototypical ("M is a very prototypical or representative member of his math [class/club]") or unprototypical ("M is a very unprototypical or unrepresentative member of his math [class/club]"). Thus, the study incorporated a 2 (group type: compulsory vs. voluntary) x 2 (target type: prototypical vs. unprototypical) design. Nineteen participants were in the compulsory/prototypical and voluntary/unprototypical conditions, while 20 participants were in the compulsory/unprototypical and voluntary/prototypical conditions.

After reading the short vignette, participants were asked to report the extent to which they expected "M" to be a respected member of his group. Participants rated target respect on a 7-point scale ranging from 1 (very disrespected) to 7 (very respected), followed by demographic questions.

## Study 1b results

A 2 (group type) x 2 (target type) ANOVA was conducted with target respect ratings as the dependent variable. There was no effect of the interaction between group type and target type, $F(1, 74) = 1.37$, $p = .25$, and no main effect of group type, $F(1, 74) = .07$, $p = .79$. There was a statistically significant and large effect of target on respect ratings, $F(1, 74) = 40.69$, $p < .001$, $g = 1.46$. Participants inferred the target to be more respected when the target was framed as a prototypical member of his group ($M = 5.28$, $SD = 1.12$) rather than an unprototypical member of his group ($M = 3.74$, $SD = .99$). Fig 2 displays these results with 95% confidence intervals for the group means. A sensitivity power analysis suggested that we had 80% power to detect effects as small as $d = .76$ for the interaction between group type and target type and $d = .64$ for the main effects.

## Study 1 discussion

The results of these two experiments provide initial support for a lay theory of groups consistent with Heider's [2] concept of balance. In order to maintain consistency in perceptions, people hold a lay theory that respected group members are prototypical group members and that prototypical group members are respected group members. Individuals use known levels of intragroup respect to infer group members' levels of prototypicality and use known levels of prototypicality to infer group members' levels of intragroup respect. They appear to do this regardless of whether the group membership is voluntary or compulsory, although there was a slight but non-significant trend wherein the gap between inferred prototypicality/respect

following manipulated respect/prototypicality is larger in voluntary vs. compulsory groups. This may be a product of slightly stronger group cohesion being inferred from voluntary membership versus compulsory membership and in Studies 2a and 2b, we specifically examine the possibility that group cohesion moderates these inferences.

## Studies 2a and 2b

Cohesion is an important barometer in perceptions of groups relative to loose collections of individuals, and cohesion may be used to make inferences about the existing prototype-respect relationship. In particular, we expect that because perceptual accentuation occurs in more highly cohesive groups, the relationship between prototypicality and respect will be accentuated to a greater degree in cohesive groups. Under the condition of Heider's [2] balance theory (see Fig 1) if a group were perceived as cohesive, prototypicality and intragroup respect would need to be viewed reciprocally—that is greater intragroup respect would lead to an inference of greater prototypicality and vice versa. If a group were perceived as incohesive then making an inference of intragroup respect from prototypicality or making an inference of prototypicality from intragroup respect would reflect a less balanced state. Study 2a examines lay inferences of prototypicality from intragroup respect within highly cohesive versus incohesive groups and Study 2b examines lay inferences of intragroup respect from prototypicality within highly cohesive versus incohesive groups.

### Study 2a method

**Participants.** One hundred and twenty-six students (69 women, 56 men) from Simon Fraser University participated in the current online study in exchange for course credit. One participant did not identify as either male or female. Participants ranged in age from 17 to 26 years ($M$ = 19.51, $SD$ = 1.52). The sample included participants identifying as East Asian (n = 59), White (n = 35), South Asian (n = 20), and 12 participants who identified with other ethnicities.

**Design and procedure.** All participants began by reading a description of prototypicality, which was defined as how representative a member is of their group. The full description can be found at https://osf.io/wxm5r/. Participants were then randomly assigned to one of four experimental conditions. Participants read a description of a group, which either emphasized the high cohesiveness of the group (e.g., "Members of Group A tend to act as a single unit. This group is highly organized. . .") or the low cohesiveness of the group (e.g., "Members of Group A rarely act as a single unit. This group is loosely organized. . ."). After completing a measure of perceived group cohesiveness as a manipulation check, participants read a statement about a member of the group, "Joe", who was either described as a respected member (e.g., "He is a respected member") or a non-respected member (e.g., "He is not a respected member"). Thus, the study incorporated a 2 (group type: cohesive vs. incohesive) x 2 (target type: respected vs. disrespected) design. Twenty-eight participants were in the cohesive/respect condition, 32 in the cohesive/disrespected condition, 32 in the incohesive/respected condition, and 34 in the incohesive/disrespected condition. We collected as much data as was feasible during the term of data collection, which was constrained by availability of participants and maximum allotments determined by the department.

After reading the short vignette, participants were asked to rate the extent that the target is a respected member of the group, which acted as a manipulation check, and then participants rated the extent that the target was perceived to be a prototypical member of the group.

**Measures.** All items for all measures used are provided in a permanent repository available here: https://osf.io/wxm5r/. We assessed group cohesiveness, intragroup respect, and prototypicality.

We measured group cohesiveness using a 14-item entitativity scale (e.g., How strongly bonded do you think that members of Group A are to their group?) adapted from Rydell and McConnell [6]. Items were rated from 1 to 7 and averaged to create a total score with higher scores indicating greater group cohesiveness ($\alpha = .96$).

Intragroup respect includes liking [12], prestige [13], competence [12], and status [14, 15]. Incorporating these aspects, we created a 6-item intragroup respect scale (e.g., "Joe is well liked by other members of group A"). Items were responded to on a scale from 1, "strongly disagree" to 5, "strongly agree" and we averaged items to create a total score with higher scores indicating greater perceived respect ($\alpha = .78$).

Prototypicality was measured using 6 items (e.g., "Joe is a very prototypical member of Group A") rated on a scale from 1, "strongly disagree" to 5, "strongly agree" and we averaged items to create a total score with higher scores indicating more perceived prototypicality ($\alpha = .73$).

## Study 2a results

We conducted two independent *t*-tests to determine if our manipulations had their intended effects. Those in the cohesive condition believed Group A to be more cohesive ($M = 5.18$, $SD = .89$) than those in the non-cohesive condition ($M = 2.75$, $SD = .98$), $t(124) = 14.50$, $p < .001$, $g = 2.59$. Additionally, those in the respect condition believed Joe to be more respected ($M = 3.57$, $SD = .48$) than those in the disrespected condition ($M = 2.21$, $SD = .51$), $t(124) = 15.45$, $p < .001$, $g = 2.74$.

To determine whether prototypicality is inferred from intragroup respect and whether this inference is dependent on the perceived cohesiveness of groups, we conducted a 2 (group type) x 2 (target type) ANOVA with perceived prototypicality of the target as the dependent variable. Results revealed a main effect of target respect, $F(1, 122) = 96.53$, $p < .001$, which was qualified by an interaction between target respect and group cohesion, $F(1, 122) = 4.43$, $p = .04$. Participants expected the target to be more prototypical when the target was framed as a respected member of his group ($M = 3.41$, $SD = .56$) rather than a disrespected member of his group ($M = 2.44$, $SD = .56$), $g = 1.73$. This effect was larger when the group was framed as cohesive ($g = 2.41$) than when the group was framed as incohesive ($g = 1.27$). Fig 3 shows this interaction. A sensitivity power analysis suggested that we had 80% power to detect effects as small as $d = .60$ for the interaction between group type and target type and $d = .50$ for the main effects.

## Study 2b method

**Participants.**   One hundred and twenty-eight students (75 women, 52 men) from Simon Fraser University participated in the current study in exchange for course credit. One

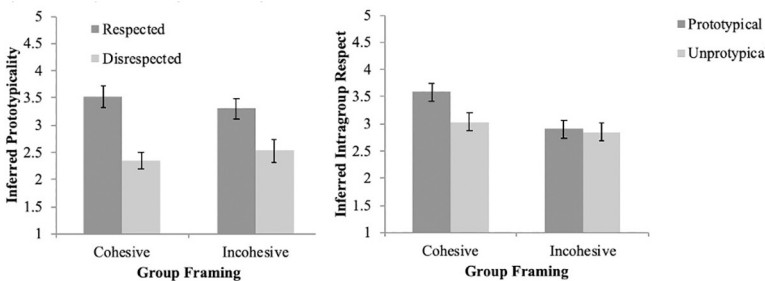

**Fig 3.  Inferred intragroup respect and inferred prototypicality as a function of target prototypicality or target intragroup respect.**

participant did not identify as either male or female. Participants ranged in age from 17 to 33 years ($M$ = 19.64, $SD$ = 1.94). The sample included participants identifying as East Asian (n = 56), White (n = 47), South Asian (n = 10), Black (n = 1), First Nations (n = 1), and 13 participants who identified with other ethnicities.

**Design and procedure.** This study was identical to 2a except "Joe" was either described as a prototypical member (e.g., "He is a prototypical member") or a non-prototypical member (e.g., "He is not a prototypical member"). Thus, the study incorporated a 2 (Group type: cohesive vs. incohesive) x 2 (Target type: prototypical vs. unprototypical) design. Thirty-one participants were in the cohesive/prototypical condition, 31 in the cohesive/unprototypical condition, 32 in the incohesive/prototypical condition, and 34 in the incohesive/unprototypical condition.

After reading the short vignette, participants were then asked to rate the extent that the target is a prototypical member of the group, which acted as a manipulation check, and then participants rated the extent that the target was perceived to be a respected member of the group.

The measures for group cohesion ($\alpha$ = .96), prototypicality ($\alpha$ = .73), and respect ($\alpha$ = .75) were the same as used in Study 2a.

## Study 2b results

We conducted two independent $t$-tests to determine if our manipulations had their intended effects. Those in the cohesive condition believed Group A to be more cohesive ($M$ = 5.32, $SD$ = .71) than those in the incohesive condition ($M$ = 2.61, $SD$ = .83), $t$ (126) = 19.91, $p < .001$, $g$ = 3.50. Additionally, those in the prototypical condition believed Joe to be more prototypical ($M$ = 3.56, $SD$ = .53) than those in the unprototypical condition ($M$ = 2.52, $SD$ = .50), $t$ (126) = 11.45, $p < .001$, $g$ = 2.02.

To determine whether intragroup respect is inferred from group member prototypicality and whether this inference is dependent on the perceived cohesiveness of groups, we conducted a 2 (group type: cohesive vs. incohesive) x 2 (target type: prototypical vs. unprototypical) ANOVA with perceived intragroup respect of the target as the dependent variable. Results revealed statistically significant main effects of target prototypicality, $F$ (1, 124) = 12.79, $p <$ .001, and group cohesion, $F$ (1, 124) = 26.38, $p < .001$, which was qualified by an interaction between target prototypicality and group cohesion, $F$ (1, 124) = 9.03, $p$ = .003. Only in the cohesive group condition did participants infer the target to be more respected when the target was framed as a prototypical member of his group ($M$ = 3.59, $SD$ = .47) rather than an unprototypical member of his group ($M$ = 3.03, $SD$ = .47), $g$ = 1.19, $t$ (60) = 4.65, $p < .001$. In the incohesive group condition, target prototypicality did not alter inferences about intragroup respect of the target, $t$ (64) = .41, $p$ = .69. These results are depicted in Fig 3. A sensitivity power analysis suggested that we had 80% power to detect effects as small as $d$ = .60 for the interaction between group type and target type and $d$ = .50 for the main effects.

## Study 2a and 2b discussion

Results from Studies 2a and 2b replicate the finding from Studies 1a and 1b, suggesting that people infer prototypicality of group members from whether that member is respected by other ingroup members and people infer intragroup respect of group members from whether they are prototypical group members. The results extend these findings by suggesting that the extent of these inferences depend on perceived cohesion of groups. While individuals inferred prototypicality from intragroup respect regardless of the cohesion of the group (left panel of Fig 3), these inferences were relatively stronger in the cohesive group, as predicted. Furthermore, our participants *only* inferred intragroup respect from prototypicality within cohesive

groups (right panel of Fig 3). This is consistent with our prediction that perceptual accentuation should increase the perceived relationship between prototypicality and intragroup respect within cohesive versus incohesive groups because it leads to perceptual harmony. Because these studies were conducted between 2008 and 2010 before preregistration had become standard practice, we opted to run a preregistered direct replication of these two studies in 2020 using higher powered designs.

## Studies 2c and 2d

Studies 2c and 2d attempted to directly replicate the results from Studies 2a and 2b respectively using a higher powered sample. The preregistration is available here: https://osf.io/4nm27.

### Study 2c method

**Participants.**   We recruited 422 participants (252 men, 165 women, 3 non-binary, 2 who identified with alternative labels) from Mechanical Turk using the CloudResearch platform [16]. Participants were compensated $0.36 for less than five minutes of their time. Participants ranged in age from 18 to 74 years ($M = 38.62$, $SD = 12.26$). The sample included participants identifying as White (n = 247), Black/African American (n = 132), American Indian/Alaskan Native (n = 6), Asian (n = 40), Hispanic/Latino/Spanish (n = 14), and Other (n = 4).

**Design and procedure.**   The design and procedures were identical to Study 2a. One-hundred and eight participants were in the cohesive/respected condition, 106 in the cohesive/disrespected condition, 105 in the incohesive/respected condition, and 103 in the incohesive/disrespected condition. Data was collected in line with our preregistration, and a sensitivity analysis suggested we had 95% power to detect interaction effects as small as $d = .40$ and main effects as small as $d = .50$.

**Measures.**   The same measures from Study 2a were used with the exception that for the group cohesion manipulation check we included only three items. We made these item selections by factor analyzing the combined data for the cohesion items from Studies 2a and 2b using principal axis factoring and specifying a single factor solution. We selected the three items with the highest factor loadings. Items 10 ("How committed do you think the members of Group A are to their group"), 11 ("How invested do you think the members of Group A are in their group"), and 12 ("How strongly bonded do you think that members of Group A are to their group") were selected on this basis and this was completed before any data were collected for Study 2c. The measures for group cohesion ($\alpha = .96$), prototypicality ($\alpha = .76$), and respect ($\alpha = .82$) all showed good internal consistency. All items for all measures used are provided in a permanent repository available here: https://osf.io/wxm5r/.

### Study 2c results

We conducted two independent $t$-tests to determine if our manipulations had their intended effects. Those in the cohesive condition believed Group A to be more cohesive ($M = 6.13$, $SD = .74$) than those in the non-cohesive condition ($M = 3.97$, $SD = 2.07$), $t (419) = 14.37$, $p < .001$, $g = 1.40$. Additionally, those in the respect condition believed Joe to be more respected ($M = 3.62$, $SD = .72$) than those in the disrespected condition ($M = 2.41$, $SD = .82$), $t (420) = 16.13$, $p < .001$, $g = 1.57$.

To determine whether prototypicality is inferred from intragroup respect and whether this inference is dependent on the perceived cohesiveness of groups, we conducted a 2 (group type) x 2 (target type) ANOVA with perceived prototypicality of the target as the dependent variable. There was a main effect of target respect, $F (1, 418) = 101.58$, $p < .001$, which was qualified by an interaction between target respect and group cohesion, $F (1, 418) = 17.46$, $p <$

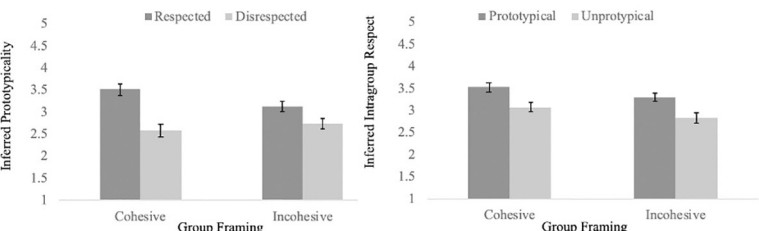

**Fig 4. Inferred intragroup respect and inferred prototypicality as a function of target prototypicality or target intragroup respect in high powered replication studies.**

.001. Participants expected the target to be more prototypical when the target was framed as a respected member of his group ($M = 3.32$, $SD = .69$) rather than a disrespected member of his group ($M = 2.65$, $SD = .71$), $g = 0.96$. This effect was larger when the group was framed as cohesive ($g = 1.30$) than when the group was framed as incohesive ($g = .61$). Fig 4 shows this interaction.

## Study 2d method

**Participants.** We recruited 427 participants (223 men, 202 women, 2 non-binary) from Mechanical Turk using the CloudResearch platform [16]. Participants were compensated $0.36 for less than five minutes of their time. Participants ranged in age from 18 to 78 years ($M = 38.66$, $SD = 12.82$). The sample included participants identifying as White (n = 283), Black/African American (n = 94), American Indian/Alaskan Native (n = 7), Asian (n = 50), Native Hawaiian/Pacific Islander (n = 5), Hispanic/Latino/Spanish (n = 32), and Other (n = 4). All participants who completed Study 2c were blocked from completing this study.

**Design and procedure.** The design and procedure was identical to Study 2b. One-hundred and nine participants were in the cohesive/prototypical condition, 105 in the cohesive/unprototypical condition, 105 in the incohesive/prototypical condition, and 108 in the incohesive/unprototypical condition. Data was collected in line with our preregistration, and a sensitivity analysis suggested we have 95% power to detect interaction effects as small as $d = .40$ and main effects as small as $d = .50$.

**Measures.** The same measures from Study 2b were used with the exception that for the group cohesion manipulation check we included only the same three items as used in Study 2c. The measures for group cohesion ($\alpha = .97$), prototypicality ($\alpha = .82$), and respect ($\alpha = .65$) all showed adequate internal consistency. All items for all measures used are provided in a permanent repository available here: https://osf.io/wxm5r/.

## Study 2d results

We conducted two independent *t*-tests to determine if our manipulations had their intended effects. Those in the cohesive condition believed Group A to be more cohesive ($M = 6.06$, $SD = .95$) than those in the non-cohesive condition ($M = 3.27$, $SD = 1.85$), $t (425) = 19.60$, $p < .001$, $g = 1.90$. Additionally, those in the prototypicality condition believed Joe to be more prototypical ($M = 3.77$, $SD = .71$) than those in the unprototypical condition ($M = 2.45$, $SD = .66$), $t (425) = 19.81$, $p < .001$, $g = 1.93$.

To determine whether intragroup respect is inferred from prototypicality and whether this inference is dependent on the perceived cohesiveness of groups, we conducted a 2 (group type) x 2 (target type) ANOVA with perceived intragroup respect of the target as the dependent variable. There was a main effect of cohesion, $F (1, 423) = 19.49$, $p < .001$ and there was a

main effect of target respect, $F(1, 423) = 76.04$, $p < .001$. These main effects were not qualified by an interaction, $F(1, 423) = 0.035$, $p = .85$. Participants expected the target to be more respected when the target group was framed as cohesive ($M = 3.30$, $SD = .62$) versus incohesive ($M = 3.06$, $SD = .57$), $g = .40$, and expected the target to be more respected when the target was framed as a prototypical member of his group ($M = 3.41$, $SD = .53$) rather than an unprototypical member of his group ($M = 2.95$, $SD = .59$), $g = 0.82$. Fig 4 demonstrates these effects.

### Study 2c and 2d discussion

Results from Studies 2c and 2d are mixed. The results from Study 2c replicated the interaction effect from Study 2a, indicating that individuals make inferences about target prototypicality from intragroup respect and this effect is accentuated when the group is perceived as cohesive (left panel of Fig 4). This is consistent with our prediction that perceptual accentuation should increase the perceived relationship between prototypicality and intragroup respect within cohesive versus incohesive groups because it leads to greater perceptual harmony. However, the results from Study 2d did not replicate the results from Study 2b. In contrast to Study 2b, the results of Study 2d indicated that participants inferred intragroup respect from target prototypicality regardless of the perceived cohesion of the group (right panel of Fig 4), in contrast to our predictions from Heider's [2] balance theory.

  Failed replications should not automatically be viewed as evidence that the original finding was a false positive [17]. Evidence for or against hypotheses must be examined holistically and with a realistic understanding of probability in mind. While failed replications can occur for many reasons, it's notable that it is very unlikely that a set of studies should yield exclusively statistically significant results [18, 19]. A series of studies with reasonable levels of power (e.g., 80%), and that are examining a hypothetically true effect, should expect one of five studies to be non-significant by chance alone. As Lakens & Etz [18] suggest, mixed results more accurately reflect reality and random variation is often a valid explanation for a single null result in a series of otherwise statistically significant findings. However, one notable difference between Study 2b and Study 2d is that the cohesion manipulation had a noticeably stronger effect on the manipulation check in 2b ($g = 3.50$) relative to 2d ($g = 1.90$), which may have enhanced the accentuation in the former.

## Study 3

Study 3 extends our findings from Study 2 by examining whether individuals make inferences about group cohesiveness based upon the perceived relationship between respect and prototypicality, which is postulated under our theoretical balance triangle depicted in Fig 1.

### Method

  **Participants.**   Seventy-seven participants (53 women, 24 men) from Simon Fraser University participated in the current online study in exchange for course credit. Participants ranged in age from 18 to 48 years ($M = 20.56$, $SD = 3.71$). The sample included participants identifying as East Asian (n = 43), White (n = 14), South Asian (n = 14), Black (n = 2), and 4 participants who identified with other ethnicities. Consistent with prior studies, we collected as much data as was feasible during the term of data collection, which was constrained by availability of participants and maximum allotments determined by the department.

  **Procedure and measures.**   All participants began by reading a description of prototypicality, which defined it as how representative a member is of their group. Participants were then randomly assigned to one of three experimental conditions. In each condition, participants began by viewing an image of 5 members of "Group A", each with his corresponding level of

intragroup respect and prototypicality (see Fig 5). The faces are available open source (CC-BY license) from Faceresearch.org [20]. The following vignette was then read by all participants.

> *These five individuals make up 'Group A.' The bars directly below each member represent the amount that each member is prototypical of Group A and the amount that each member is respected within Group A. Please take a few moments to look over the prototypicality and respect levels of each group member.*

In the "positive" condition the image of the group members reflected a strong positive association between intragroup respect and prototypicality (i.e., members who were more prototypical were also more respected). In the "neutral" condition, there was no relationship between intragroup respect and prototypicality, and in the "negative" condition there was a negative association between intragroup respect and prototypicality (i.e., members who were more prototypical were also less respected). To ensure the manipulated associations were noticed and salient, we explicitly reminded participants of the relationship (e.g., in the positive condition we stated, "Notice that the most prototypical member is also the most respected member and the least prototypical member is the least respected. Overall, the more prototypical a member is the more respected he is").

Following the experimental manipulation, participants completed the following manipulation check, "Suppose we tell you about another member of Group A. Based on the patterns above, how respected would you expect him to be if his level of prototypicality was 2?" Participants responded from 0–5, the same scale that the group members' respect and prototypicality were shown in the images. Participants could also select an option that reflected not enough information available to make the decision, which should be selected more frequently in the neutral condition. Because the relationship between prototypicality and intragroup respect was intentionally manipulated, there was one predictable answer for each condition based upon the relationship shown in the images. Finally, participants completed the same measure of group cohesiveness used in Studies 2a and 2b.

## Results

We examined how well participants were able to select the predictable value for target respect based upon a value of "2" for prototypicality. In the positive condition (n = 20), 75% of participants selected the predictable value that corresponded to a strong positive association between prototypicality and respect. In the negative condition (n = 31), 71% of participants selected the predictable value that corresponded to a strong negative association between prototypicality and respect. In the neutral condition (n = 26), any value would have been plausible since there was no association between variables. Forty-two percent stated that there was not enough information to choose and the remainders were dispersed across alternative responses. After eliminating the "not enough information" responses, the means for the positive (*M* = 2.20, *SD* = .62) and negative (*M* = 3.93, *SD* = .47) responses reflected the values of "2" and "4" respectively—the correct values. The mean of the neutral condition (*M* = 2.87, *SD* = .83) approximately reflected the midpoint of the scale at 2.5. Additionally, the standard deviation of responses is consistent with the neutral condition being more ambiguous even after removing the participants who selected the "not enough information" option. In short, participants understood the presented relationships between respect and prototypicality.

We tested whether Group A was seen as more cohesive when respect and prototypicality were positively associated than when they were unrelated or negatively associated by conducting two independent samples *t*-tests (see Fig 6). Group A was seen as significantly more

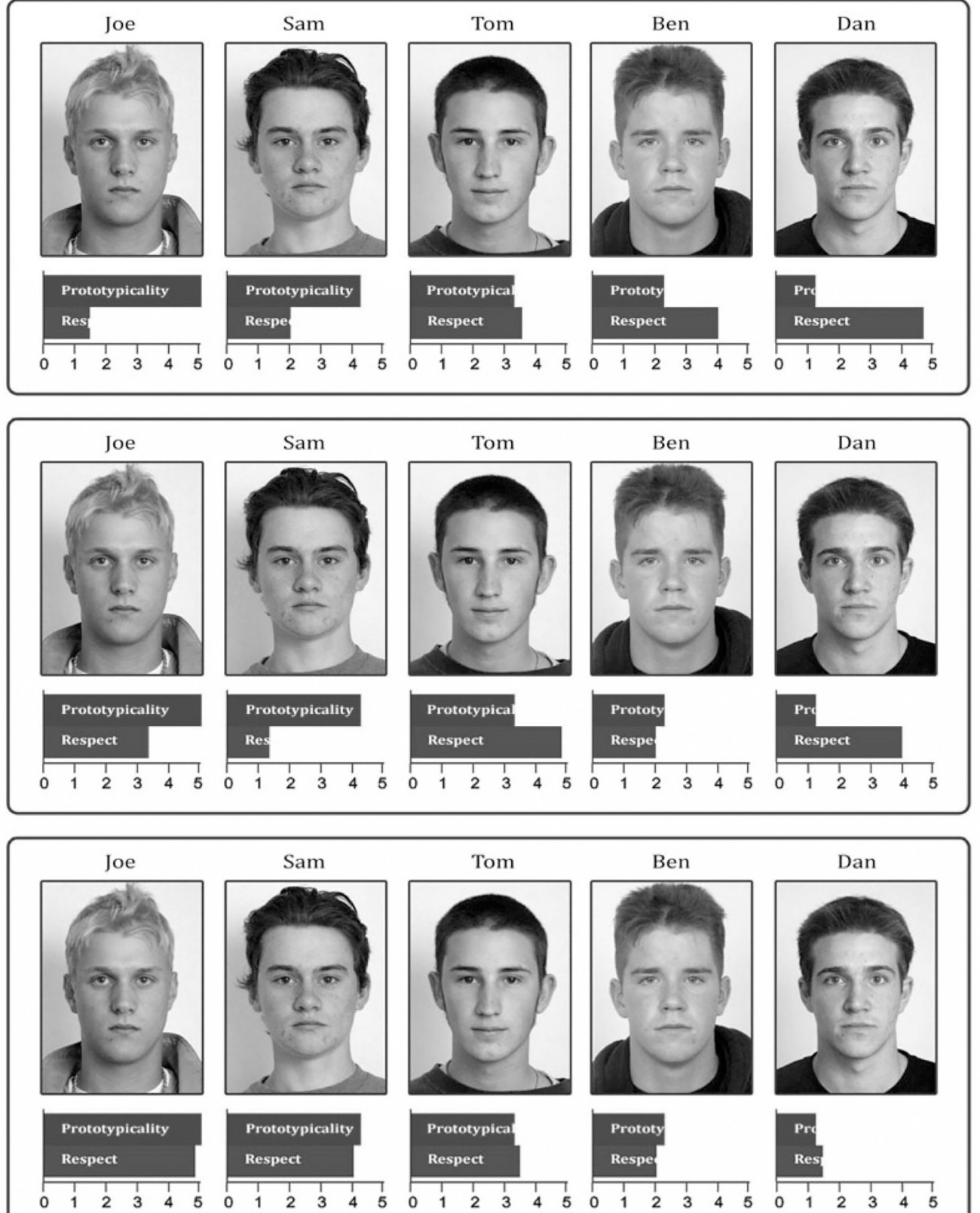

**Fig 5. Conditions reflecting negative (top), neutral (middle), and positive (bottom) associations between prototypicality and respect.**

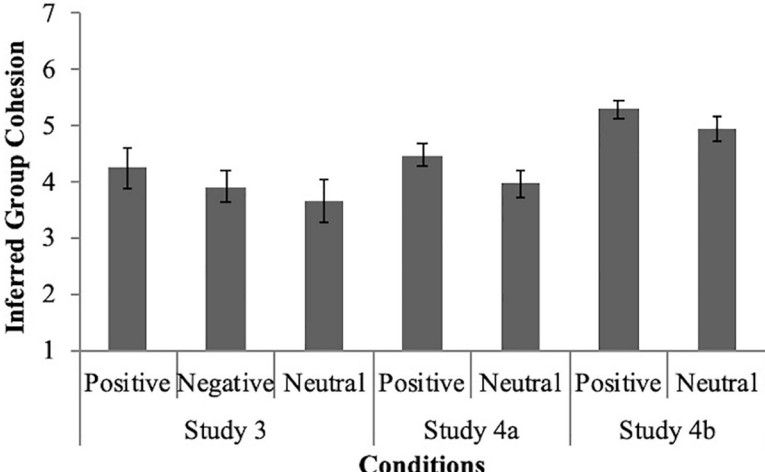

**Fig 6. Inferred group cohesiveness as a function of the relationship between prototypicality and intragroup respect.**

cohesive in the positive condition ($M = 4.25$, $SD = .84$) compared to the neutral condition ($M = 3.64$, $SD = .99$), $t(44) = 2.21$, $p = .03$, $g = .66$. Additionally, Group A was not seen as significantly more cohesive in the positive condition compared to the negative condition ($M = 3.91$, $SD = .79$), $t(49) = 1.47$, $p = .15$, $g = .42$., though examining the effect size does still suggest a trend in the predicted direction. In balance terms, perceived cohesion should be higher in the positive condition than the negative condition because making a stronger inference of cohesion would reflect perceptual harmony (three positive relations) in the positive condition. In contrast, making a reduced inference of cohesion would reflect perceptual harmony (two negative relations and one positive relation) in the negative condition. Group A was also not seen as significantly more cohesive in the negative condition compared to the neutral condition, $t(55) = 1.15$, $p = .26$, $g = .30$, which is consistent with predictions from our balance triangle. A sensitivity power analysis suggested that we had 80% power to detect effects as small as $d = .85$ for the comparison between the positive and neutral conditions and $d = .82$ for the comparison between the positive and negative conditions.

## Discussion

For Heider [2], entities "form a unit when they are perceived as belonging together" (p. 176). This study provides the first evidence that perception of traits being in balance (i.e., respect and prototypicality) implies group cohesiveness to lay individuals. Individuals appear to infer greater group cohesiveness when traits are in perceptual harmony (i.e., when prototypicality and intragroup respect are seen as belonging together). When traits are in perceptual harmony, inferences about group cohesion are strongest. A case with negative correlations (where the prototypical members are the least respected) is an unlikely real-world scenario. We suspect that individuals perceived this relationship as odd and had difficulty inferring group cohesion in such a way that reflected perceptual harmony because of the ambiguous scenario that has little real world parallel. The ambiguity in the scenario led participants to select more neutral response options on the cohesion scale. That 55% of respondents in the negative condition had group cohesion scores clustered around the midpoint (between 3.5 and 4.5) is supportive of this view. In contrast, only 35% of respondents in the positive or neutral conditions inferred group cohesion scores that clustered around the midpoint. While the balance principle may generally

be used in order to anticipate patterns of relations among entities, participants may abandon the use of the balance principle when it no longer makes intuitive sense to apply it [5].

## Study 4a and 4b

In Study 4a we conducted a replication of Study 3 with an increased sample size. Because of the ambiguity in making inferences of group cohesion in the negative condition in Study 3, in Study 4a we chose to focus on the positive condition, which clearly reflects the case of perceptual harmony. We also used a more realistic group paradigm rather than the generic categorizing of "Group A" and we extended the analysis to determine whether ingroup members, rather than only outsiders, also use a folk theory that reflects Heider's balance theory. In Study 4b we conducted a high-powered and preregistered replication of Study 4a. The preregistration is available here: https://osf.io/4nm27.

### 4a method

**Participants.** One hundred and twelve participants (81 women, 31 men) from Simon Fraser University participated in the current online study in exchange for course credit. Consistent with the prior studies, we collected as much data as was feasible during the term of data collection, which was constrained by availability of participants and maximum allotments determined by the department. Participants ranged in age from 17 to 35 years ($M = 20.15$, $SD = 2.77$) but one participant did not report her age. The sample included participants identifying as East Asian (n = 59), White (n = 37), South Asian (n = 15), and 1 participant who identified with another ethnicity.

**Procedures and measures.** Participants were randomly assigned to one of two experimental manipulations. In each condition, participants began by reading the following short vignette:

*Please take a few moments and imagine you belong to a social group called the FunFinders that organizes weekly get-togethers (e.g., movies, dinner, and concerts, etc.) for students outside of the university. Like all groups, some members of this group contribute in more positive ways to the group and are most associated with what it means to be a 'FunFinder.'*

For the positive condition (n = 55), the vignette continued:

*As a member of this group you notice that these 'prototypical' FunFinders members get much more respect than the ones that don't contribute much and do not really embody the group. So, some Funfinders members are more respected than other members, and their level of respect appears to be related to how prototypical they are.*

For the neutral condition (n = 57), the vignette read:

*As a member of this group you notice that these 'prototypical' FunFinders members don't get any more respect than the ones that don't contribute much and do not really embody the group. So, some Funfinders members are more respected than other members, but their level of respect appears not to be related to how prototypical they are.*

Participants then completed the same group cohesiveness measure from Study 3 and basic demographic questions. No manipulation check was included in Study 4 since we had already demonstrated the efficacy of this experimental procedure in Study 3.

### 4a results

To determine whether group cohesiveness is inferred from the relationship between group member prototypicality and intragroup respect, we conducted an independent *t*-test. Results revealed that participants in the positive condition (n = 55) inferred greater group cohesion (*M* = 4.48, *SD* = .73) than participants in the neutral condition (n = 57, *M* = 3.97, *SD* = .96), *t* (110) = 3.14, *p* = .002, *g* = .60. Fig 6 displays these results with 95% confidence intervals for the mean estimates. A sensitivity power analysis suggested that we had 80% power to detect effects as small as *d* = .53 for this statistical test.

### 4b method

**Participants.** We recruited 213 participants (133 men, 80 women) from Mechanical Turk using the CloudResearch platform [16]. Participants were compensated $0.36 for less than five minutes of their time. Participants ranged in age from 20 to 76 years (*M* = 35.12, *SD* = 10.69). The sample included participants identifying as White (n = 146), Black/African American (n = 45), American Indian/Alaskan Native (n = 3), Asian (n = 14), Hispanic/Latino/Spanish (n = 16), Hawaiian/Pacific Islander (n = 1), and Other (n = 1). All participants who completed either Study 2c or 2d were blocked from completing this study.

**Procedures and measures.** The procedures and measures were identical to Study 4a. The group cohesion measure showed good internal consistency (α = .95). All materials are available here: https://osf.io/wxm5r/.

### 4b results

To determine whether group cohesiveness is inferred from the relationship between group member prototypicality and intragroup respect, we conducted an independent *t*-test. Results revealed that participants in the positive condition (n = 108) inferred greater group cohesion (*M* = 5.29, *SD* = .82) than participants in the neutral condition (n = 105, *M* = 4.94, *SD* = 1.17), *t* (210) = 2.58, *p* = .01, *g* = 0.35. Fig 6 displays these results with 95% confidence intervals for the means. A sensitivity power analysis suggested that we had 95% power to detect effects as small as *d* = .50 for this statistical test.

### 4a and 4b discussion

One perception that is important in determining whether a group is cohesive is whether the prototypical members, those members that best represent cumulative positive attributes of the membership [1, 10], are respected by other group members. In Study 4a we replicate the results from Study 3, which suggested that participants do infer greater group cohesion when intragroup respect and prototypicality of a target are framed as positively correlated relative to when these variables are framed as unrelated. In Study 4b, we replicated this result again in a high powered and preregistered sample. The perception of group cohesiveness from the relationship between prototypicality and intragroup respect reflects the ability of lay individuals to make inferences that reflect a consistent, coherent, and simple representation of groups [2, 3].

### *P*-curve

When sets of studies report true effects, the reported *p*-values from these studies will form a distribution of statistically significant *p*-values that is positively skewed—there will be a greater number of small *p*-values (e.g., .01) relative to large *p*-values (e.g., .04). In contrast, non-existent effects lead to either flat distributions with a similar number of small and large *p*-values or to negatively skewed distributions with a greater number of large *p*-values relative to small *p*-

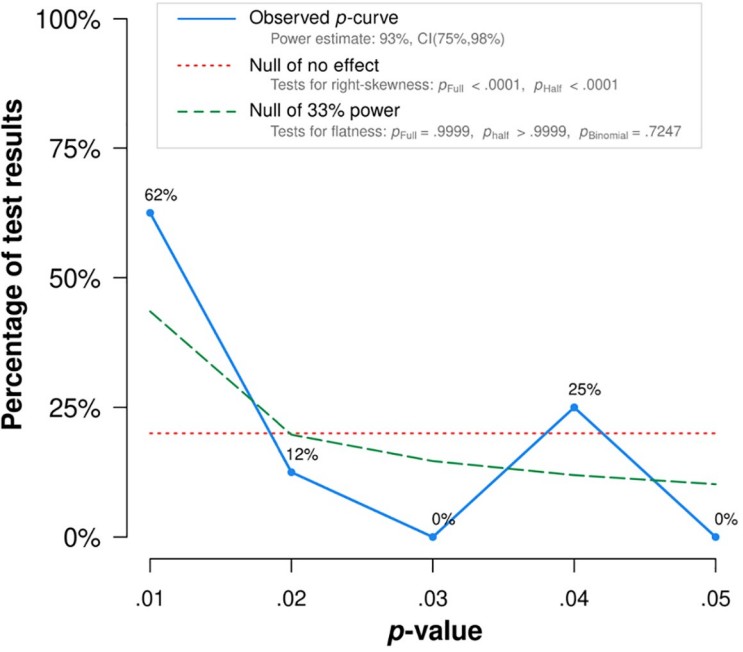

**Fig 7. Full *p*-curve indicating evidential value.**

values. *P*-curve [21] computes two statistical tests to determine whether there is evidential value in a set of reported results. The first examines whether the positive skew is statistically significant, which indicates evidential value. In our case, both the full *p*-curve ($Z$ = -5.82, $p$ < .001) and half *p*-curve ($Z$ = -6.41, $p$ < .001) are statistically significant, indicating evidential value. Half *p*-curve utilizes only the distribution of *p*-values below .025, which leads to a more robust test of evidential value [21]. The second test examines whether the curve is flatter than a hypothetical curve with 33% power, which would indicate that the evidential value is inadequate. In our case, our curve is not flatter than one with 33% power in the full ($Z$ = 3.66, $p$ = .999) or the half *p*-curve ($Z$ = 5.96, $p$ > .999), which indicates that the evidential value of this set of studies is adequate. Fig 7 displays the full *p*-curve and Table 1 displays the summary information used in the *p*-curve and summarizes the results of our studies following the recommended format of *p*-curve [21].

## General discussion

Heider [2] suggested that people are motivated to form a harmonious perception of social stimuli in order to maintain affective and cognitive consistency, or "balance". We extended this notion to the case of groups, wherein individuals infer characteristics of group members (e.g., prototypicality and intragroup respect) in a consistent manner. Evidence in Studies 1a and 1b suggested that individuals make inferences about group member prototypicality from intragroup respect and that individuals make inferences about group member respect from member prototypicality. These inferences maintain a consistent and harmonious representation of group members.

According to Heider [2], entities "form a unit when they are perceived as belonging together" (p. 176), and our Studies 2a and 2b demonstrate that, when referencing cohesive groups, individuals infer a stronger relationship between prototypicality and intragroup

**Table 1. Summary of results.**

| Study | Prediction of Interest | Study Design | Key Result | Quoted Text with Result | Result |
|---|---|---|---|---|---|
| 1a | "…we begin by examining whether lay individuals infer the prototypicality of members of groups from knowledge of the extent to which a member is respected by other ingroup members…" | 2 (respected vs. disrespected) x 2 (voluntary vs. compulsory group) | Simple Effect | "There was a statistically significant and large effect of target on inferred prototypicality, $F(1, 76) = 44.83$, $p < .001$, $g = 1.52$. Participants inferred higher prototypicality of the target when the target was framed as a respected member of his group ($M = 4.78$, $SD = 1.35$) rather than a disrespected member of his group ($M = 2.80$, $SD = 1.26$)." | $F(1, 76) = 44.83$ |
| 1b | "…and infer intragroup respect of members of groups from knowledge of how prototypical a group member is…" | 2 (prototypical vs. non-prototypical) x 2 (voluntary vs. compulsory group) | Simple Effect | "There was a statistically significant and large effect of target on respect ratings, $F(1, 74) = 40.69$, $p < .001$, $g = 1.46$. Participants inferred the target to be more respected when the target was framed as a prototypical member of his group ($M = 5.28$, $SD = 1.12$) rather than an unprototypical member of his group ($M = 3.74$, $SD = .99$)." | $F(1, 74) = 40.69$ |
| 2a | "…inferences about prototypicality from intragroup respect…should be stronger within highly cohesive versus incohesive groups." | 2 (respected vs. disrespected) x 2 (cohesive vs. incohesive) (attenuated interaction) | Two-way interaction | "…an interaction between target respect and group cohesion, $F(1, 122) = 4.43$, $p = .04$. Participants expected the target to be more prototypical when the target was framed as a respected member of his group ($M = 3.41$, $SD = .56$) rather than a disrespected member of his group ($M = 2.44$, $SD = .56$), $g = 1.73$. This effect was larger when the group was framed as cohesive ($g = 2.41$) than when the group was framed as incohesive ($g = 1.27$)." | $F(1, 122) = 4.43$ |
| 2b | "…inferences about…intragroup respect from prototypicality should be stronger within highly cohesive versus incohesive groups." | 2 (prototypical vs. non-prototypical) x 2 (cohesive vs. incohesive) (attenuated interaction) | Two-way interaction | "…an interaction between target prototypicality and group cohesion, $F(1, 124) = 9.03$, $p = .003$. Only in the cohesive group condition did participants infer the target to be more respected when the target was framed as a prototypical member of his group ($M = 3.59$, $SD = .47$) rather than an unprototypical member of his group ($M = 3.03$, $SD = .47$), $g = 1.19$, $t(60) = 4.65$, $p < .001$. In the incohesive group condition, target prototypicality did not alter inferences about intragroup respect of the target, $t(64) = .41$, $p = .69$." | $F(1, 124) = 9.03$ |
| 2c | "…inferences about prototypicality from intragroup respect…should be stronger within highly cohesive versus incohesive groups." | 2 (respected vs. disrespected) x 2 (cohesive vs. incohesive) (attenuated interaction) | Two-way interaction | "an interaction between target respect and group cohesion, $F(1, 418) = 17.46$, $p < .001$. Participants expected the target to be more prototypical when the target was framed as a respected member of his group ($M = 3.32$, $SD = .69$) rather than a disrespected member of his group ($M = 2.65$, $SD = .71$), $g = 0.96$. This effect was larger when the group was framed as cohesive ($g = 1.30$) than when the group was framed as incohesive ($g = .61$)." | $F(1, 418) = 17.46$ |
| 2d | "…inferences about…intragroup respect from prototypicality should be stronger within highly cohesive versus incohesive groups." | 2 (prototypical vs. non-prototypical) x 2 (cohesive vs. incohesive) (attenuated interaction) | Two-way interaction | "These main effects were not qualified by an interaction, $F(1, 423) = 0.035$, $p = .85$." | $F(1, 423) = 0.035$ |
| 3 | "…examining whether individuals make inferences about group cohesiveness based upon the perceived relationship between respect and prototypicality, which is postulated under our theoretical balance triangle depicted in *Fig 1*." | 3-cell (positive correlation, no correlation, negative correlation) | Positive vs. No correlation | "Group A was seen as significantly more cohesive in the positive condition ($M = 4.25$, $SD = .84$) compared to the neutral condition ($M = 3.64$, $SD = .99$), $t(44) = 2.21$, $p = .03$, $g = .66$." | $t(44) = 2.21$ |

*(Continued)*

**Table 1.** (Continued)

| Study | Prediction of Interest | Study Design | Key Result | Quoted Text with Result | Result |
|---|---|---|---|---|---|
| 4a | "In Study 4 we conducted a replication of Study 3 with an increased sample size. Because of the ambiguity in making inferences of group cohesion in the negative condition in Study 3, in Study 4 we chose to focus on the positive condition, which clearly reflects the case of perceptual harmony." | 2-cell (positive vs. no correlation) | Difference of means | "Results revealed that participants in the positive condition (n = 55) inferred greater group cohesion ($M = 4.48$, $SD = .73$) than participants in the neutral condition (n = 57, $M = 3.97$, $SD = .96$), $t (110) = 3.14$, $p = .002$, $g = .60$." | $t (110) = 3.14$ |
| 4b | "In Study 4b we conducted a high-powered and preregistered replication of Study 4a." | 2-cell (positive vs. no correlation) | Difference of means | "Results revealed that participants in the positive condition (n = 108) inferred greater group cohesion ($M = 5.29$, $SD = .82$) than participants in the neutral condition (n = 105, $M = 4.94$, $SD = 1.17$), $t (210) = 2.58$, $p = .01$, $g = 0.35$." | $t (210) = 2.58$ |

respect than when referencing incohesive groups. In high powered and preregistered replication attempts (2c & 2d), we successfully replicated 2a but failed to replicate the predicted interaction effect in 2b; participants did infer respect from prototypicality, but they did so to similar degrees regardless of group cohesiveness. Results of Studies 3 and 4a demonstrate that inferences about group cohesion based upon target prototypicality and target respect are altered in order to maintain perceptual harmony. In a high powered and preregistered study (4b) we successfully replicated this effect. Ultimately, this collection of findings supports the existence of a network of associations between prototypicality, intragroup respect, and group cohesion that serves the purpose of making inferences that maintain perceptual harmony [1, 22, 23].

Common sense psychologies, such as the one our participants used, are useful because they generally help individuals make accurate inferences [24]. Social categorization theorists have investigated the associations between prototypicality and intragroup respect among group members, finding that protypical members are, in fact, liked more by their fellow ingroup members, are more popular within the group, and that this relationship is stronger within more cohesive groups [9]. For example, in an analysis of an Australian football team, more prototypical teammates were more socially desired as potential teammates for a future team [8] and prototypical group leaders receive more support from ingroup members than non-prototypical leaders [25]. This can be viewed as a broader application of the ingroup favoritism principle [26]. Liking is selectively generalized to fellow ingroup members, and because prototypical members are the best representation of the ingroup identity, they subsequently receive a greater degree of liking than less prototypical ingroup members. Our results suggest that lay individuals perceive and infer these relationships, which allows them to make accurate predictions about groups and corresponding group members.

Self-categorization theory is concerned with explaining "how people come to think, feel, and act as a psychological group. . ." [27, p. 403]. Both prototypicality and intragroup respect are important variables within SCT. Explaining who becomes a prototypical group member is central to the theory, and self-categorization researchers have attempted to explain who becomes a prototypical group member by way of the meta-contrast principle—prototypical group members are as similar as possible to other ingroup members and as different as possible to outgroup members [27, 28]. These prototypical members best represent the distinct group identity, are subsequenty liked more than less prototypical members, become a valid source of information for other group members and are consequently viewed as leaders worthy of respect [27, 29–31]. For self-categorization theorists, prototypicality "is the yardstick of

group life" [30, p. 129] and prototypicality is the cause of intragroup respect and group-serving behaviors [32].

However, the inferred relationship between prototypicality and intragroup respect is bidirectional. If respect acts as a signal for prototypicality, as our results suggest, then engaging in behaviors that elicit respect from group members could increase one's perceived prototypicality regardless of meta-contrast within the specific ingroup-outgroup comparison. Individuals, especially new group members, may use intragroup respect as the marker for determining who should be listened to and modeled within the group. Follower behavior may initially be driven by perceptions of intragroup respect, rather than prototypicality, and respected group members may be able to alter group prototypes without being seen as norm-violators. In part, this could be because intragroup respect may be easier to determine for group members than prototypicality, and thus a more practical guide for follower behavior, since it can be identified based upon observation of other group members behaving reverantly toward the highly respected members. Respect can also be communicated directly, whereas prototypicality may not be stated explicitly. In some cases individuals may identify and mirror the norms and values of perceived respected group members, who will also be the more prototypical members. This might be especially important for new members to groups who must determine how to behave appropriately.

Because both intragroup respect and prototypicality can influence group behavior and are highly reciprocal, when attempting to make claims about the role of either it becomes important to control for the other in experiments. For example, when male students are presented with false feedback about how prototypical they are of the category "men", high identifiers report more negative affect when told they are unprototypical [33]. Since perceptions of prototypicality and respect are confounded, it is plausible that it may actually be participants' perceptions that they will not be respected by other men that drives their negative affect when told they are unprototypical. Jetten, Spears, and Manstead [34] suggested that prototypical group members are more inclined to exhibit ingroup bias in response to distinctiveness threat. It is also plausible to interpret their findings as *respected group members* are more inclined to exhibit ingroup bias in response to distinctiveness threat for the purpose of not losing respect, and their corresponding position, within the group. Because respect and prototypicality involve reciprocal inference, anytime intragroup respect (or prototypicality) is manipulated or measured, so too is prototypicality (or respect).

Finally, our results may have implications for understanding discrimination of non-prototypical job applicants. For example, in hiring of women relative to men, hiring committees will have knowledge about who is and is not prototypical in their field. For example, if a hiring board recognizes that only 4.8% of Fortune 500 CEOs are women [35], this may be indicative to board members that women are not prototypical in that role and that men are prototypical. As our results suggest, this knowledge of prototypicality may lead to inferences that women are less respected CEOs than men. This inference may have negative repercussions for equitable hiring. Jackson, Esses, and Burris [36] used mock letters of recommendation to experimentally manipulate the extent that a job candidate was perceived as respected. Applicants who were described as respected received higher ratings than those that were not described this way, and this was especially pronounced for a high-status job. If, as Jackson et al. suggest, perceptions of respect causally impact hiring evaluations, and if perceived prototypicality induces inferences of intragroup respect, then it seems likely that inferences of prototypicality have a distal and causal impact on gender disparities in hiring.

The lower relative respect that women may receive relative to men in certain careers may also have other personal ramifications. Respect matters because it acts as an indicator of worth and the formation of meaningful relationships—two core motives of social life [37]. Intragroup

respect is linked to self-esteem, health, well-being, and treatment by peers [37, 38]. Knowing the extent to which group members are respected provides information regarding their intragroup behavior. For example, evidence suggests that individuals who are respected by their ingroup peers are more likely to engage in group-serving behaviors, such as voluntary activities that enhance the reputation of the ingroup [13, 15]. Most importantly, our studies demonstrate that individuals do appear to use a practical lay theory to guide perceptions of group members that generally follows the Heiderian principle of balance.

## Acknowledgments

We wish to thank Sophia Johl and Ramsay Malange for their help with data collection.

## Author Contributions

**Conceptualization:** Joshua D. Wright, L. James Climenhage, Michael T. Schmitt, Nyla R. Branscombe.

**Data curation:** Joshua D. Wright, L. James Climenhage.

**Formal analysis:** Joshua D. Wright.

**Funding acquisition:** Michael T. Schmitt.

**Investigation:** Joshua D. Wright, L. James Climenhage, Michael T. Schmitt.

**Methodology:** Joshua D. Wright, L. James Climenhage, Michael T. Schmitt, Nyla R. Branscombe.

**Project administration:** Joshua D. Wright, L. James Climenhage, Michael T. Schmitt.

**Resources:** Joshua D. Wright, L. James Climenhage, Michael T. Schmitt.

**Software:** Joshua D. Wright, L. James Climenhage, Michael T. Schmitt.

**Validation:** Joshua D. Wright.

**Visualization:** Joshua D. Wright.

**Writing – original draft:** Joshua D. Wright.

**Writing – review & editing:** Joshua D. Wright, L. James Climenhage, Michael T. Schmitt, Nyla R. Branscombe.

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
