## [Decision Letter · Decision Letter 0]

8 Oct 2020

PONE-D-20-23353

Perceptual Harmony in Judgments of Group Prototypicality and Intragroup Respect

PLOS ONE

Dear Dr. Wright,

Thank you for submitting your manuscript to PLOS ONE. After careful consideration, we feel that it has merit but does not fully meet PLOS ONE’s publication criteria as it currently stands. Therefore, we invite you to submit a revised version of the manuscript that addresses the points raised during the review process.

We look forward to receiving your revised manuscript.

Kind regards,

Angela Sutan, PhD Economics

Academic Editor

PLOS ONE

Journal Requirements:

3. We note that Figure 5 includes an image of participants in the study. 

Reviewers' comments:

Reviewer's Responses to Questions

**Comments to the Author**

1. Is the manuscript technically sound, and do the data support the conclusions?

Reviewer #1: Yes

Reviewer #2: Yes

2. Has the statistical analysis been performed appropriately and rigorously? 

Reviewer #1: Yes

Reviewer #2: Yes

3. Have the authors made all data underlying the findings in their manuscript fully available?

Reviewer #1: No

Reviewer #2: Yes

4. Is the manuscript presented in an intelligible fashion and written in standard English?

Reviewer #1: Yes

Reviewer #2: Yes

5. Review Comments to the Author

Reviewer #1: Interesting article, that will need some improvement in the ways the numerous experiments are presented. First a table may help the reader be prepared to digest the experiment. In the current form the read through of the paper feel repetitive and a table may provide a clear view of the results and may save some explanation in full text. Improvement on the presentation of the result to avoid copy pasted of the same paragraph may help.

The motivation in the introduction section could be made more salient by explaining why the topic is important (like presented in the discussion section).

Another aspect is the non-result that was not expected by the experiment 2b) could the author formulate some additional test on distribution (maybe) or some slight explanations for the reader or to guide future researcher. Overall, some distributional test maybe appreciated, as they mentioned latter (experiment 3) unfamiliar situation lead to noisy results.

The justification for removing scales in experiments 2c and 2d is unclear, even if done correctly, why was it needed? Some experiment are repetition with larger sample and make the reader wonder if they could be pooled (can be tested) with the small initial sample and test if the results are maintained.

Last some typos and consistency should be checked (ex: p24 number format, p34 typos, etc.). Figure 1 is hard to view prone to be low resolution as the writing is too small.

Reviewer #2: The authors tell us a good story.

The paper is well written, and the studies are presented in a logical manner.

I enjoyed reading it very much!

Most of my objections were answered by the manipulation checks.

I have some minor questions:

- You always mention in your studies participants’ ethnic identification. In your studies participants mainly identify as East Asian. Do you believe it influences their answers? In your preregistered studies the identifications are different. Do you believe it has an impact?

- You often provide a link to some materials, it is often a general link: https://osf.io/wxm5r/ would it be possible to provide a specific link instead because it is tough to find the mentioned material regarding the quantity of material provided.

- Is there an explanation why some items are responded on a 5-points Likert scale and some other ones on a 7-points Likert scale?

- You didn’t provide enough explanations about the fact that the results from Study 2d did not replicate the results from Study 2b.

- In Study 3 why did you choose male faces. Do you expect the same results with female faces?

- Page 28 (eee Simonsohn, Simmons, and Nelson (2015) for additional explanation)- I assume “see” instead of “eee”.

6. PLOS authors have the option to publish the peer review history of their article (what does this mean?). If published, this will include your full peer review and any attached files.

Reviewer #1: **Yes: **Gassmann Xavier

Reviewer #2: No

---

## [Author Response · Author response to Decision Letter 0]

24 Nov 2020

Journal Requirements:

We have updated the formatting of the manuscript to be aligned with the PLOS ONE guidelines, including altering the heading levels to have no more than three levels and changing the font size of levels, changing the figure formatting, changing the formatting of the author information, eliminating the running head, adding continuous line numbers, and adopting Vancouver style citations. 

We have added the ethics statement to the Methods section for Study 1A and have removed it from all other sections. It now states, “All studies received ethics approval from the Research Ethics Board at Simon Fraser University.”

3. We note that Figure 5 includes an image of participants in the study. 

Figure 5 does not include images of participants in the study. Images used in Figure 5 are under a CC-BY license and available open source from Faceresearch.org. This is indicated on page 21. 

Reviewer 1:

Interesting article, that will need some improvement in the ways the numerous experiments are presented. First a table may help the reader be prepared to digest the experiment. In the current form the read through of the paper feel repetitive and a table may provide a clear view of the results and may save some explanation in full text. Improvement on the presentation of the result to avoid copy pasted of the same paragraph may help.

All nine studies build upon each other and have small differences between them and these details are important for understanding the studies and for potential replication of the studies in the future. Where possible we have avoided repetition. For example, when describing Study 4b we state, “The procedures and measures were identical to Study 4a” and in Study 1b we state, “This study was identical to 1a except…” We have done this throughout the paper. We have now provided a new table (Table 1), which serves two purposes. First, it provides a summary of all nine studies, including the hypothesis, the design, and the key result of each. Second, it follows guidelines for reporting p-curves. 

The motivation in the introduction section could be made more salient by explaining why the topic is important (like presented in the discussion section).

We have added additional material to the introduction (see page 4 & 6) to explain the practical use of the principle of balance among people but still retain most discussion of implications in the discussion, where implications of theory are generally noted. The purpose of our studies, as indicated in the introduction, is to test Heider’s balance theory. Testing theory is the hallmark of science. We wish to avoid repetition by duplicating the material that is already in the discussion section.

Another aspect is the non-result that was not expected by the experiment 2b) could the author formulate some additional test on distribution (maybe) or some slight explanations for the reader or to guide future researcher. Overall, some distributional test maybe appreciated, as they mentioned latter (experiment 3) unfamiliar situation lead to noisy results.

Failed replications should not automatically be viewed as evidence that the original finding was a false positive (Open Science Collaboration, 2015). Evidence for or against hypotheses must be examined holistically and with a realistic understanding of probability in mind. While failed replications can occur for many reasons, it’s notable that it is very unlikely that a set of studies should yield exclusively statistically significant results (Lakens & Etz, 2017; Schimmack, 2012). A series of studies with reasonable levels of power (e.g., 80%), and that are examining a hypothetically true effect, should expect one of five studies to be non-significant by chance alone. As Lakens & Etz (2017) suggest, mixed results more accurately reflect reality and random variation is often a valid explanation for a single null result in a series of otherwise statistically significant findings.

We have included this on page 21. We have added the following: “However, one notable difference between Study 2b and Study 2d is that the cohesion manipulation had a noticeably stronger effect on the manipulation check in 2b (g = 3.50) relative to 2d (g = 1.90), which may have enhanced the accentuation in the former.”

The justification for removing scales in experiments 2c and 2d is unclear, even if done correctly, why was it needed? Some experiment are repetition with larger sample and make the reader wonder if they could be pooled (can be tested) with the small initial sample and test if the results are maintained.

This decision was made prior to collecting any data, as indicated in the preregistration. The decision was made based upon the length of the experiment and the funds that were available. We wanted to reduce the amount of time participants needed to spend completing the study in order to bring the length of the study to a level that corresponded to a reimbursement amount of the national minimum wage in the United States. Because the scale was used only as a manipulation check we didn’t feel that 14 items were necessary to determine whether a manipulation had worked or not.

Pooling a replication and the original study would defeat the purpose of the replication. Direct replications are necessary for determining the robustness of effects. Pooling is a form of p-hacking, which we desire to avoid. It is also clear that the results are maintained since the studies (with exception of 2b) all replicated in independent samples conducted 12 years after the original studies. These studies were conducted on different samples in different contexts, which should bolster confidence in the findings. 

Last some typos and consistency should be checked (ex: p24 number format, p34 typos, etc.). Figure 1 is hard to view prone to be low resolution as the writing is too small.

We have removed these typos and we have created a higher resolution version.

Reviewer #2: The authors tell us a good story.

The paper is well written, and the studies are presented in a logical manner.

I enjoyed reading it very much!

Most of my objections were answered by the manipulation checks.

I have some minor questions:

- You always mention in your studies participants’ ethnic identification. In your studies participants mainly identify as East Asian. Do you believe it influences their answers? In your preregistered studies the identifications are different. Do you believe it has an impact?

No, we do not have any reason to believe that being East Asian influences the effects seen in our studies. Study 2c and 4b were successful replications with more diverse samples than the earlier studies with high East Asian proportions. There are no demographic specifications for effects within Heider’s Balance Theory. 

- You often provide a link to some materials, it is often a general link: https://osf.io/wxm5r/ would it be possible to provide a specific link instead because it is tough to find the mentioned material regarding the quantity of material provided.

Unfortunately, the OSF does not have the functionality to provide separate links for each item. However, we have provided clearer labels for downloadable material. Within the link, we have created folders for each study labeled according to their presentation in the paper. Now, if a researcher were interested in Study 3, that research could identify the relevant data and code by selecting the folder labeled “Study 3”. 

- Is there an explanation why some items are responded on a 5-points Likert scale and some other ones on a 7-points Likert scale?

Yes, we followed the scale as designed by the creators of the scale. If the original paper from which a scale was drawn utilized a 7-point scale then we did the same. If the original paper from which a scale was drawn utilized a 5-point scale, then we did the same. 

- You didn’t provide enough explanations about the fact that the results from Study 2d did not replicate the results from Study 2b.

We have expanded on this in the paper. Notably, with approximately 80% power across a set of studies, researchers should expect 20% of studies to be a null result by chance alone. We have also added a comment about the manipulation in 2d having a weaker effect. I have reproduced the new section from pages 21 below:

“Failed replications should not automatically be viewed as evidence that the original finding was a false positive [17]. Evidence for or against hypotheses must be examined holistically and with a realistic understanding of probability in mind. While failed replications can occur for many reasons, it’s notable that it is very unlikely that a set of studies should yield exclusively statistically significant results [18, 19]. A series of studies with reasonable levels of power (e.g., 80%), and that are examining a hypothetically true effect, should expect one of five studies to be non-significant by chance alone. As Lakens & Etz [18] suggest, mixed results more accurately reflect reality and random variation is often a valid explanation for a single null result in a series of otherwise statistically significant findings. However, one notable difference between Study 2b and Study 2d is that the cohesion manipulation had a noticeably stronger effect on the manipulation check in 2b (g = 3.50) relative to 2d (g = 1.90), which may have enhanced the accentuation in the former.” 

- In Study 3 why did you choose male faces. Do you expect the same results with female faces?

The faces only serve the purpose of increasing the external validity of the task by creating “real” group members to be evaluated. The active ingredient in the manipulation is the level of prototypicality and respect labeled beneath each photo. We suspect that inferences would be the same regardless of gender of targets in the photos or whether there were no photos at all. Afterall, we did run study 4a and 4b without the need for photos and the results obtained were the same. 

- Page 28 (eee Simonsohn, Simmons, and Nelson (2015) for additional explanation)- I assume “see” instead of “eee”.

This has been corrected.

---

## [Editor Report · Decision Letter 1]

27 Nov 2020

Perceptual Harmony in Judgments of Group Prototypicality and Intragroup Respect

PONE-D-20-23353R1

Dear Dr. Wright,

We’re pleased to inform you that your manuscript has been judged scientifically suitable for publication and will be formally accepted for publication once it meets all outstanding technical requirements.

Kind regards,

Angela Sutan, PhD Economics

Academic Editor

PLOS ONE
---

## [Editor Report · Acceptance letter]

4 Dec 2020

PONE-D-20-23353R1 

Perceptual harmony in judgments of group prototypicality and intragroup respect 

Dear Dr. Wright:

I'm pleased to inform you that your manuscript has been deemed suitable for publication in PLOS ONE. Congratulations! Your manuscript is now with our production department. 

Kind regards, 

on behalf of

Dr. Angela Sutan 

Academic Editor

PLOS ONE